# Tunable anomalous Hall conductivity through volume-wise magnetic competition in a topological kagome magnet

Z. Guguchia [1,2]*, J.A.T. Verezhak[1], D.J. Gawryluk [3], S.S. Tsirkin [4], J.-X. Yin[2], I. Belopolski[2], H. Zhou[5,6], G. Simutis[1], S.-S. Zhang [2], T.A. Cochran[2], G. Chang [2], E. Pomjakushina [3], L. Keller[7], Z. Skrzeczkowska[3], Q. Wang[8], H.C. Lei [8], R. Khasanov[1], A. Amato [1], S. Jia[5,6], T. Neupert [4], H. Luetkens[1]* & M.Z. Hasan [2]*

Magnetic topological phases of quantum matter are an emerging frontier in physics and material science. Along these lines, several kagome magnets have appeared as the most promising platforms. Here, we explore magnetic correlations in the kagome magnet $Co_3Sn_2S_2$. Using muon spin-rotation, we present evidence for competing magnetic orders in the kagome lattice of this compound. Our results show that while the sample exhibits an out-of-plane ferromagnetic ground state, an in-plane antiferromagnetic state appears at temperatures above 90 K, eventually attaining a volume fraction of 80% around 170 K, before reaching a non-magnetic state. Strikingly, the reduction of the anomalous Hall conductivity (AHC) above 90 K linearly follows the disappearance of the volume fraction of the ferromagnetic state. We further show that the competition of these magnetic phases is tunable through applying either an external magnetic field or hydrostatic pressure. Our results taken together suggest the thermal and quantum tuning of Berry curvature induced AHC via external tuning of magnetic order.

[1] Laboratory for Muon Spin Spectroscopy, Paul Scherrer Institute, CH-5232 Villigen PSI, Switzerland. [2] Laboratory for Topological Quantum Matter and Spectroscopy, Department of Physics, Princeton University, Princeton, NJ 08544, USA. [3] Laboratory for Multiscale Materials Experiments, Paul Scherrer Institut, 5232 Villigen PSI, Switzerland. [4] Department of Physics, University of Zürich, Winterthurerstrasse 190, Zurich, Switzerland. [5] International Center for Quantum Materials and School of Physics, Peking University, Beijing, China. [6] CAS Center for Excellence in Topological Quantum Computation, University of Chinese Academy of Science, Beijing, China. [7] Laboratory for Neutron Scattering, Paul Scherrer Institut, CH-5232 Villigen PSI, Switzerland. [8] Department of Physics and Beijing Key Laboratory of Opto-electronic Functional Materials and Micro-nano Devices, Renmin University of China, Beijing, China. *email: zurab.guguchia@psi.ch; hubertus.luetkens@psi.ch; mzhasan@princeton.edu

The kagome lattice is a two-dimensional pattern of corner-sharing triangles. With this unusual symmetry and the associated geometrical frustration, the kagome lattice can host peculiar states including flat bands[1], Dirac fermions[2,3] and spin liquid phases[4,5]. In particular, magnetic kagome materials offer a fertile ground to study emergent behaviours resulting from the interplay between unconventional magnetism[6] and band topology[7–9]. Recently, transition-metal-based kagome magnets[1–5,10–14] are emerging as outstanding candidates for such studies, as they feature both large Berry curvature fields and unusual magnetic tunability. In this family, the kagome magnet $Co_3Sn_2S_2$ is found to exhibit both a large anomalous Hall effect and anomalous Hall angle, and is identified as a promising Weyl semimetal candidate[10,12,15,16]. However, despite knowing the magnetic ground state is ferromagnetic below $T_C = 177$ K[17] with spins aligned along the $c$-axis[10,12,18] (see Fig. 1a, b) there is no report of its magnetic tunability or phase diagram, and its interplay with the topological band structure.

Here we use high-resolution $\mu$SR to systematically characterize the phase diagram, uncovering another intriguing in-plane anti-ferromagnetic phase. The magnetic competition between these two phases is further found to be highly tunable via applying either pressure[19–22] or magnetic field. Combined with first-principles calculations, we discover that the tunable magnetic correlation plays a key role in determining the giant anomalous Hall transport.

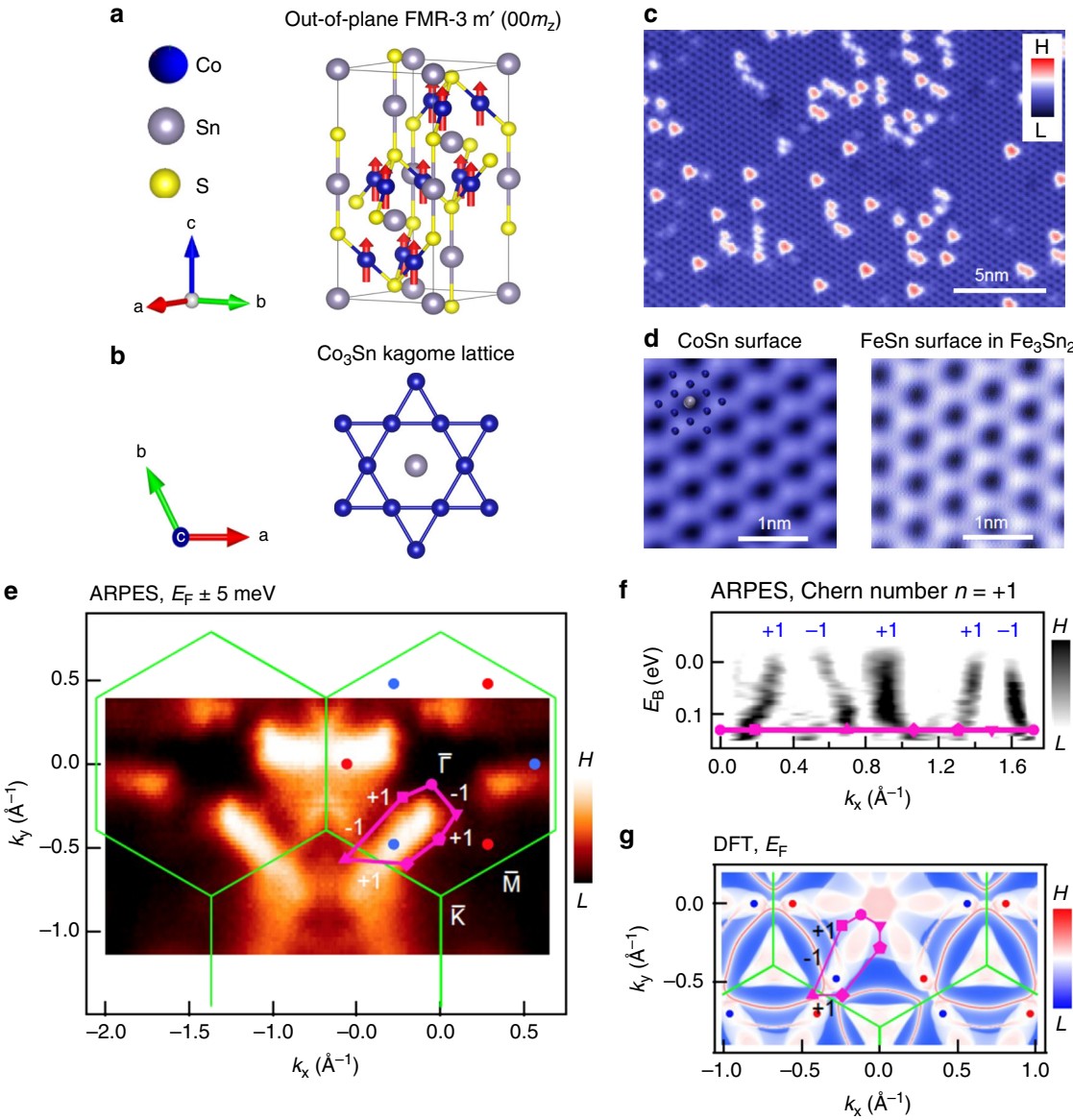

**Fig. 1 Topological ground state of the kagome system $Co_3Sn_2S_2$. a** Magnetic structure of $Co_3Sn_2S_2$, showing a ferromagnetic ground state with spins on Co atoms aligned along the $c$-axis. **b** Kagome lattice structure of the $Co_3Sn$ layer. **c** Topographic image of the CoSn surface. **d** A zoom-in image of the CoSn surface (left) that shows similar morphology with the FeSn surface (right) in $Fe_3Sn_2$. The inset illustrates the possible atomic assignment of the kagome lattice. Data are taken at the tunnelling junction: $V = 50$ mV, $I = 0.8$ nA, $T = 4.2$ K. **e** Fermi surface of $Co_3Sn_2S_2$ acquired using angle-resolved photoemission spectroscopy (ARPES) at temperature $T = 22$ K with surface Brillouin zone (green lines) determined from the crystal structure; predicted topological band crossing points (positive Chern number: blue dots, negative Chern number: red dots) from first-principles calculation; and a closed surface momentum-space loop (purple quadrilateral). **f** Energy-momentum cut along the purple loop, indicating an unconventional odd number of band crossings. **g** Calculated (momentum-resolved) surface density of states at $E_F$ for $Co_3Sn_2S_2$.

## Results

**Topological ground state.** $Co_3Sn_2S_2$ has a layered crystal structure with a CoSn kagome lattice (Fig. 1a, b). Cleaving at cryogenic temperatures often reveals Sn and S terminated surfaces as demonstrated by our previous scanning tunnelling microscopy (STM) study[1]. In addition to these two dominant surfaces, we also very rarely found CoSn surfaces (Fig. 1c, left panel of d) which lies under the S surface. An enlarged view of this surface reveals a similar morphology similar to the FeSn surface in $Fe_3Sn_2$ at the atomic level[2], both of which are consistent with the transition-metal based kagome lattice structure as seen in the STM images in Fig. 1d. Having confirmed the fundamental kagome lattice in the crystal structure, we further characterize its electronic structure by angle-resolved photoemission spectroscopy (ARPES). We study the Fermi surface at a temperature of 22 K and observe regions of broad spectral weight suggesting bulk states, interspersed with sharper features suggesting surface states (Fig. 1e). Based on fundamental theoretical considerations, the electronic structure of a ferromagnet generically hosts Weyl fermions in the bulk, associated with a non-zero Chern number. Furthermore, for a closed momentum-space path in the surface Brillouin zone, the net number of surface states on the path equals the net enclosed bulk Chern number[23–25]. Motivated by these considerations, we study the ARPES dispersion on a closed loop enclosing the observed bulk pocket (purple contour in Fig. 1e). We observe five crossings at the Fermi level, inconsistent with any conventional surface state band structure (Fig. 1f). We next sum up the net number of crossings on the loop, associating a +1 with left-propagating modes and a −1 with right-propagating modes. We find a net value of +1, suggesting that the loop encloses a Chern number of $n = +1$ and implying that at least one Weyl fermion lies inside the loop. To more deeply understand our result, we calculate the distribution of Weyl points from DFT and we observe that one Weyl point

projects inside our closed loop (blue dot in Fig. 1e), consistent with our observation of a Chern number of $n = +1$ using ARPES. We next calculate the entire Fermi surface and we observe that the Weyl points are connected pairwise by Fermi arc surface states, along with trivial surface state pockets enclosing the $\bar{K}$ and $\bar{K}'$ points (Fig. 1g, see also Supplementary Note 8). We can understand the crossings observed in ARPES by associating them with these surface states in DFT. Proceeding from left to right in Fig. 1f, we see that the first crossing (+1) is the Fermi arc, the next two (−1 and +1) are associated with the $\bar{K}$ point surface state, while the last two (+1 and −1) may be associated with a surface resonance arising from bulk pockets projecting near $\bar{\Gamma}$. In this way, the correspondence between ARPES and DFT allows us to identify the first chiral mode in Fig. 1f as the topological Fermi arc surface state. Our experimental observations suggest the existence of a topologically non-trivial band structure in the kagome magnet $Co_3Sn_2S_2$.

**Ferromagnetic ground state.** Having characterized the topologically non-trivial band structure, we probe the crystal and magnetic structure of $Co_3Sn_2S_2$ at base temperature (Fig. 2a, b). The crystal structure of $Co_3Sn_2S_2$ through the entire temperature range was well refined with Rietveld refinements of the raw neutron diffraction data, employing a rhombohedral lattice structure in the space group R-3m. An example of the refinement profile for the 2 K data is shown in Fig. 2a, b, and no secondary phase can be detected. We do not observe any substantial additional diffraction intensity at 2 K in comparison with the paramagnetic state. Instead, it appears that the only clearly visible effect on the Bragg peak intensities originates from the decrease in the Debye-Waller atomic displacement parameters. To estimate the upper limit of the magnetic moment at the Co sites as well as to identify the magnetic structure, we have performed a

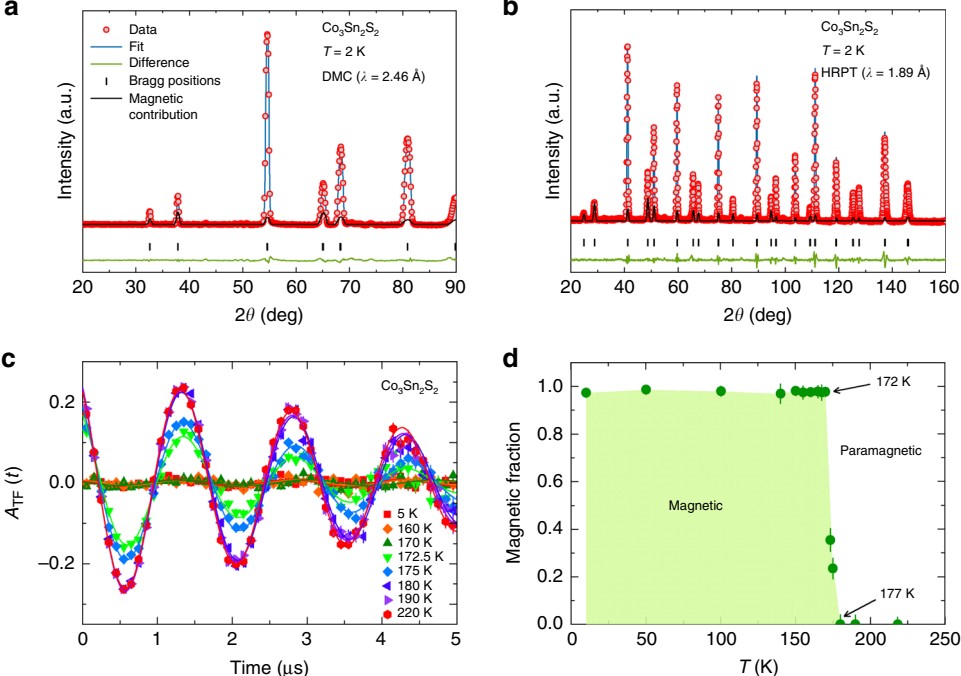

**Fig. 2 Ferromagnetic ground state and the ordered fraction versus temperature in $Co_3Sn_2S_2$. a, b** Neutron powder diffraction pattern, recorded at 2 K for the sample $Co_3Sn_2S_2$ with two different instruments. The solid black lines represent a Rietveld refinement profile. The residuals are plotted at the bottom of the figure. The solid green lines are the fitted magnetic contributions. To better visualise the magnetic peaks, the intensities are multiplied by 500 and 300 for (**a**) and (**b**), respectively. **c** The weak-TF $\mu$SR spectra, obtained above and below Curie temperature $T_C$. **d** The temperature dependence of the magnetically ordered volume fraction, extracted from the amplitude of the TF $\mu$SR spectra. The error bars represent the s.d. of the fit parameters.

group theoretical analysis, described in detail below, and a Rietveld refinement of the data. Since we know from the magnetic susceptibility data that the sample is out-of-plane ferromagnetically ordered[17], the minimal realistic model to refine at 2 K is with the R-3m' structure. By including only the ferromagnetic (FM) $z$-component in the refinement we deduce $m_z = 0.269(102) \mu_B$. By including all parameters in the refinement, as well as the atomic displacements (ADP) we obtain a magnetic moment, $\mathbf{M} = (m_x, m_y, m_z) = [0.093(80), 0.187(160), 0.255(137)] \mu_B$. Thus the upper estimate of the long range ordered magnetic moment per Co at base temperature is 0.2–0.3 $\mu_B$. Our neutron diffraction data support the $z$-axis FM order in $Co_3Sn_2S_2$ and provides an estimate for the upper bound of the ordered moment.

**Temperature dependence of magnetic fraction and magnetic structure.** Next we study the magnetism and its temperature, pressure and field dependence in $Co_3Sn_2S_2$ using the $\mu$SR technique, which serves as an extremely sensitive local probe for detecting small internal magnetic fields and ordered magnetic volume fractions in the bulk of magnetic materials. The weak transverse-field (weak-TF) $\mu$SR spectra and the temperature dependence of the magnetically ordered volume fraction for $Co_3Sn_2S_2$ are shown in Fig. 2c, d, respectively. In a weak-TF, the

amplitude of the low-frequency oscillations is proportional to the paramagnetic volume fraction. Thus, a spectrum with no oscillation corresponds to a fully ordered sample, while a spectrum with oscillation in the full asymmetry indicates that the sample is in a paramagnetic state. The weak-TF spectra below $T \sim 172$ K show negligibly small 3% oscillation amplitudes, demonstrating a fully magnetically ordered ground state. Above 172 K, the oscillation amplitude increases and reaches the full paramagnetic volume at $T_C \simeq 177$ K. The temperature-dependent magnetic fraction therefore shows a relatively sharp transition from the paramagnetic to the magnetic state with the coexistence of magnetic and paramagnetic regions in the temperature interval 172–177 K, i.e., only very close to the transition.

In order to study the detailed temperature evolution of the magnetic order parameter in $Co_3Sn_2S_2$, zero-field $\mu$SR measurements were carried out. Time spectra, recorded below (5, 170, and 175 K) and above (180 K) the magnetic ordering temperature, are shown in Fig. 3a. At $T = 180$ K, the entire sample is in the paramagnetic state as evidenced by the weak $\mu$SR depolarization and its Gaussian functional form arising from the interaction between the muon spin and randomly oriented nuclear magnetic moments[26]. At $T = 5$ K, a spontaneous muon-spin precession with a well-defined single frequency is observed, which is clearly

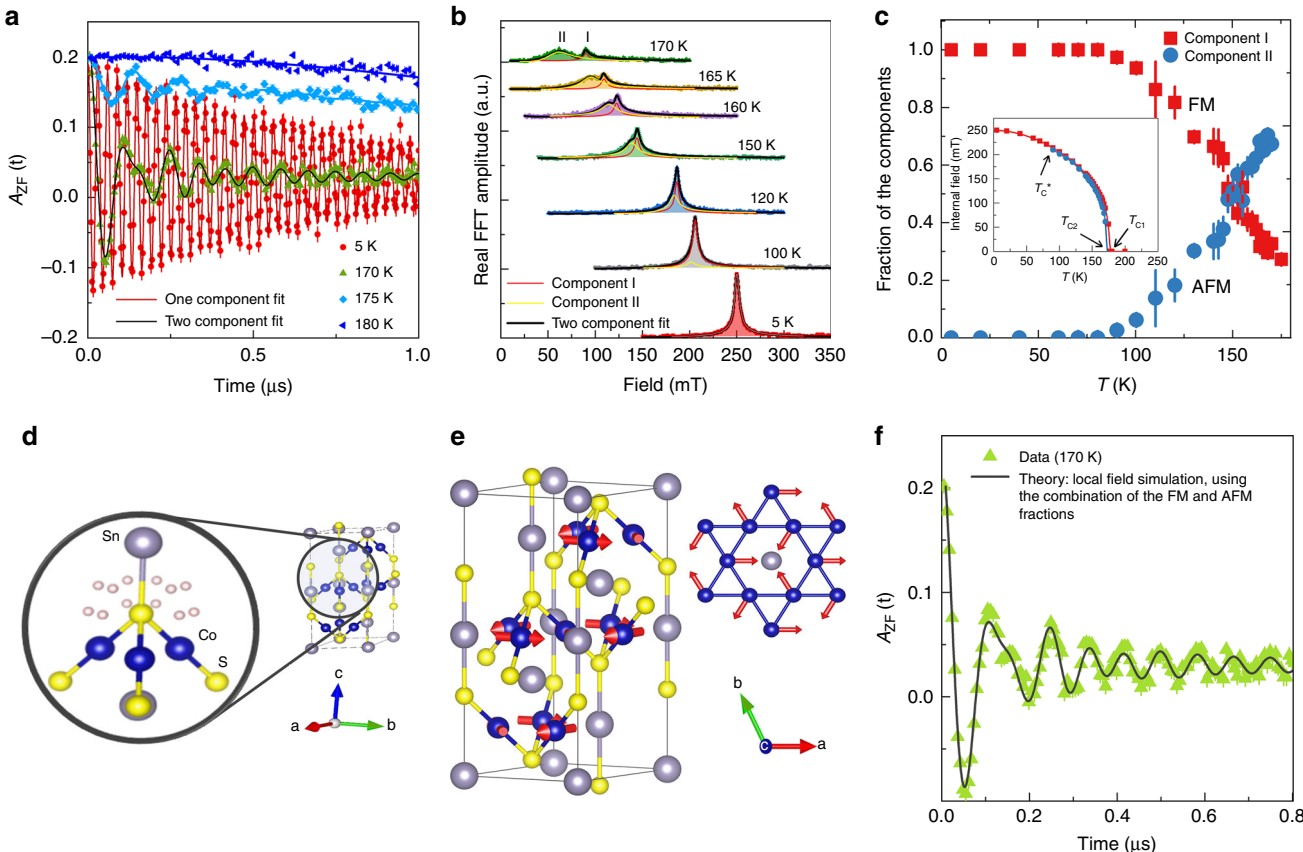

**Fig. 3 Phase separation between two distinct magnetically ordered regions in Co₃Sn₂S₂. a** Zero-field spectra, recorded at temperatures above and below $T_C$. The solid lines are the fit of the data using Eq. (2). Error bars are the s.e.m. in about $10^6$ events. The error of each bin count $n$ is given by the s.d. of $n$. The errors of each bin in $A(t)$ are then calculated by s.e. propagation. **b** Fourier transform amplitudes of the oscillating components of the $\mu$SR time spectra as a function of temperature. **c** The temperature dependences of the relative volume fractions of the two magnetically ordered regions. Inset shows the temperature dependences of the internal magnetic fields for the two components. Arrows mark the critical temperatures $T_{C1}$ and $T_{C2}$ for high frequency and low-frequency components, respectively as well as the transition temperature $T_C^*$, below which only one component signal is observed. The error bars represent the s.d. of the fit parameters. **d** Crystallographically equivalent muon stopping sites with the Wyckoff position 36i within the structure of CoSn₂S₂. **e** In-plane antiferromagnetic structure. Grey solid lines indicate the boundaries of a single unit cell of the crystal structure. **f** $\mu$SR spectrum, recorded at 170 K, is shown along with the result of the local field simulation (black solid line) at the muon stopping site, considering two distinct magnetically ordered regions with out-of-plane and in-plane magnetic configurations, respectively.

visible in the raw data (Fig. 3a). In addition, the spectrum is characterised by the low value of the transverse muon-spin depolarization rate $\lambda_T = 1.2(1)\,\mu s^{-1}$, which is a measure of the width of the static magnetic field distribution at the muon site, implying a narrow field distribution in the sample and thus a very homogeneous magnetic ground state. Our stopping site calculations and stability analysis reveal one plausible muon stopping site to be the Wyckoff position 36i (see Fig. 3d) in $Co_3Sn_2S_2$, which is consistent with the observation of only one precession frequency in the ZF-$\mu$SR spectra at base temperature [details on the muon stopping site and local field calculations can be found in the Supplementary Notes 6 and 7]. The single frequency behaviour is robust up to ~90 K, after which a second frequency begins to appear (Fig. 3a).

To better visualize this effect, we show the Fourier transform amplitudes of the oscillating components of the $\mu$SR time spectra as a function of temperature (Fig. 3b). Starting from $T = 90$ K, two distinct precession frequencies appear in the $\mu$SR spectra, which can be clearly seen and are well separated when approaching the transition temperature. The temperature dependences of the internal fields ($\mu_0 H_{int} = \omega/\gamma_\mu^{-1}$) for the two components are shown in the inset of Fig. 3c. Both order parameters show the monotonous decrease and clear separation with increasing temperature. It is important to note that the two components have slightly different transition temperatures, with the high frequency component having an onset at $T_{C1} \simeq 177$ K, and the low-frequency component having an onset of $T_{C2} \simeq 172$ K. Remarkably, the low-frequency component develops at the cost of the high frequency component, since the appearance of the low-frequency component above $T_C^* \sim 90$ K is accompanied by the reduction of the volume fraction of the high frequency one. As shown in Fig. 3c, the fraction of the high frequency component continuously decreases with increasing the temperature above $T_C^* \sim 90$ K and close to the transition the low-frequency component acquires the majority of the volume. These results suggest the presence of two distinct magnetically ordered regions in $Co_3Sn_2S_2$ at temperatures above ~90 K (see also the Supplementary Note 5). Note that the disappearance of the low-frequency component above $T_{C2} \simeq 172$ K explains the reduction of the total magnetic fraction in the temperature interval between $T_{C2} \simeq 172$ K and $T_{C1} \simeq 177$ K (see Fig. 2d).

To understand the multi-domain physics further, we investigate the possible magnetic structures in $Co_3Sn_2S_2$. According to previously reported magnetisation measurements, $Co_3Sn_2S$ exhibits out-of-plane FM ordering below $T_{C1} \simeq 177$ K. The FM structure forces the propagation vector to be $\mathbf{k} = (0,0,0)$, so we consider the maximally symmetric Shubnikov magnetic subgroups of the grey paramagnetic group R-3m1' for the $\Gamma$-point of the Brillouin zone. The decomposition of the magnetic representation for Co in Wyckoff position 9d (1/2,0,1/2) reads $\Gamma_1^+ \otimes 2\Gamma_2^+ \otimes 3\Gamma_3^+$ ($\tau_7$) with the dimensions of irreproducible representations (irreps) one, one, and two, respectively. The first two irreps result in the R-3m' and R-3m subgroups, which are the solutions of maximal symmetry with the smallest number of the refined parameters. The first subgroup R-3m' has two spin components $\mathbf{M}_{Co} = (m_x, 2m_x, m_z)$ with AFM in the plane and FM along the $c$-axis. Previous reports[18] as well as our neutron experiments suggest a $c$-axis aligned R-3m' FM structure (see Fig. 1a) (although some canting away from the $c$-axis is possible) as the magnetic ground state for $Co_3Sn_2S_2$. The second subgroup R-3m is 120° antiferromagnetic (AFM) order with the spins in ($ab$) plane (see Fig. 3e), which we will refer to as in-plane AFM order. In addition to the above symmetry considerations, we also investigate the magnetic structure using density functional theory (DFT). We find that among the two magnetic arrangements as

shown in Figs. 1a and 3e, the lowest energy configuration is FM with the magnetic moments along the $c$-axis, with magnitude ~0.35 $\mu_B$/atom. The in-plane AFM configuration with the same magnetic moment has a higher energy of ~20 meV/Co atom, relative to the $c$-axis FM order. The energy difference between AFM and FM states is found to be insensitive with the Hubbard $U$ and with the temperature variation of the lattice constants (see Supplementary Note 1 and Supplementary Fig. 1b). Given the information of the allowed magnetic structures and the muon stopping site in $Co_3Sn_2S_2$, the local fields were calculated for both out-of-plane FM and in-plane AFM spin arrangements at the single muon site with the Wyckoff position 36i (see Fig. 3d) and then fit with the fractions and moment sizes for each phase. The result of the calculation for 170 K is shown in Fig. 3f as the black dashed line, which lines up with the experimental data very well. Moreover, the temperature dependence of the moment sizes, relative fractions of two magnetic components and the relaxation rates (see Supplementary Note 7) also agree very well with the experimental data. Thus, the analysis confirms the two magnetically ordered fractions in $Co_3Sn_2S_2$ with different moment sizes: at low temperatures the out-of-plane FM structure is dominant and with increasing temperature the fraction of the in-plane AFM state grows, and becomes the dominant component at 170 K.

The schematic magnetic phase diagram and spin structures of $Co_3Sn_2S_2$ are drawn in Fig. 4a, showing the evolution of the out-of-plane FM and in-plane AFM states as a function of temperature. Our key finding is the observation of a phase separated magnetically ordered region in the temperature interval of $T_C^* < T < T_{C2}$ in the magnetic Weyl semimetal candidate $Co_3Sn_2S_2$. This conclusion is robust and model-independent, as it relies on the extreme sensitivity of $\mu$SR, which is a local probe, to ordered magnetic volume fractions in the bulk of magnetic materials. Neutron diffraction measurements provide an estimate of the upper bound of the long range ordered magnetic moment 0.2 – 0.3$\mu_B$ per Co at base temperature. Local field simulations at the muon stopping site further shed light on the possible magnetic states. For $T < T_C^* = 90$ K, the entire sample exhibits an out-of-plane FM structure. When $T \geq 90$ K, the in-plane AFM arrangement appears, and its volume fraction grows with increasing temperature and eventually dominates around 170 K, before it disappears at $T_{C2} = 172$ K. Above $T_{C2}$, the sample exhibits the small volume fraction with the out-of-plane FM order (the rest of the volume is occupied by the paramagnetic state) until $T_{C1} = 177$ K, before reaching a fully paramagnetically ordered state. According to our DFT calculations the energies of the out-of-plane FM and in-plane AFM configurations are similar. Thermodynamics may affect this intricate balance, and tip it in favour of the in-plane magnetic structure. The interplay between different magnetically ordered regions, each of which may possess distinct topological invariants, can possibly give rise to exciting physics at the magnetic domain boundaries.

The $\mu$SR observation of the presence of the magnetic phase for $T_C^* < T < T_{C2}$ in $Co_3Sn_2S_2$ is also supported by the temperature-dependent magnetic susceptibility data (see Supplementary Note 3), which shows the broad maximum within that temperature range. However, we note that the anomaly in susceptibility appears at ~125 K, which is higher than $T_C^* \sim 90$ K observed by $\mu$SR. This is most likely related to the fact that the volume fraction of the new magnetic phase is small at 90 K and gradually increases with increasing temperature. Since the microscopic $\mu$SR technique is extremely sensitive to small volume fractions, it allows it to probe the true onset temperature of the transition. Macroscopic magnetization measurements detect the transition at higher temperatures, where the magnetic phase acquires a high enough volume fraction.

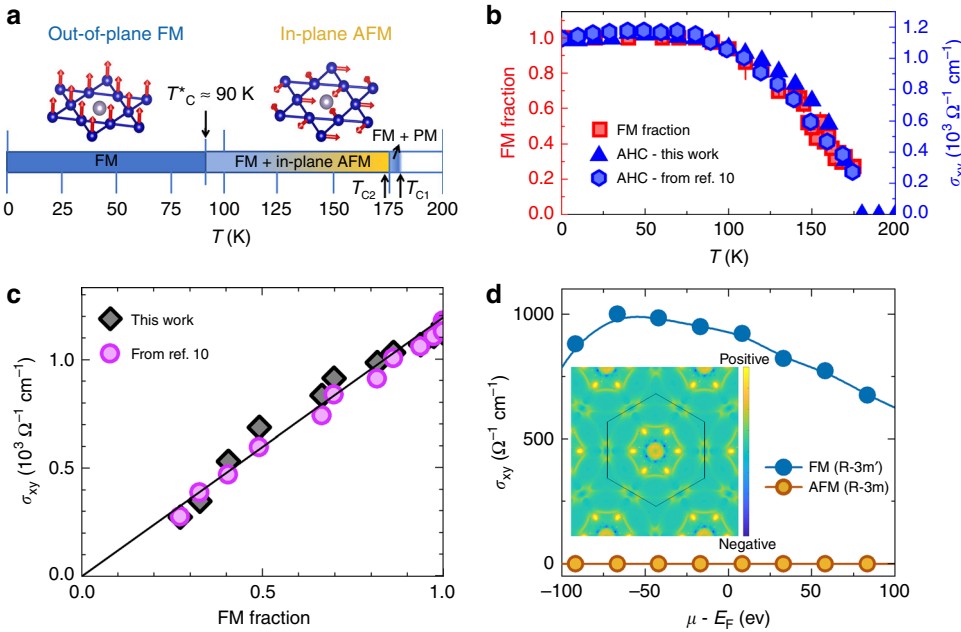

**Fig. 4 Magnetic competition driven thermal evolution of anomalous Hall conductivity. a** Schematic magnetic phase diagram as a function of temperature and spin structures of $Co_3Sn_2S_2$, i.e. the FM and the in-plane AFM structures. The arrows mark the transition temperatures $T_{C1} \simeq 177$ K, $T_{C2} \simeq 172$ K and $T_C^*$ $\simeq 90$ K. **b** The temperature dependence of the FM volume fraction and the in-plane anomalous Hall conductivity $\sigma_{xy}$. The error bars of the FM fraction represent the s.d. of the fit parameters. **c** The correlation plot of $\sigma_{xy}$ vs fraction of the ferromagnetically ordered region. The solid straight line is drawn between a hypothetical situation of the minimum (zero) and the maximum values of $\sigma_{xy}$ and the FM fraction. Data shown in solid circles are taken from ref. [10]. **d** The dependence of $\sigma_{xy}$ on the chemical potential, calculated for the out-of-plane FM and the in-plane AFM structures. The inset shows the calculated Berry curvature distribution in the BZ at the Ferromagnetic phase.

**Linear correlation between anomalous Hall and magnetic fraction.** One of the most striking effects in $Co_3Sn_2S_2$ is a large intrinsic anomalous Hall conductivity (AHC) and a giant anomalous Hall angle, due to the considerably enhanced Berry curvature arising from its band structure[10]. In order to explore the correlation between the magnetic properties found in this work and the topological aspects of $Co_3Sn_2S_2$, we compare the temperature dependence of the anomalous Hall conductivity to the temperature evolution of the volume fraction of the out-of-plane FM component (Fig. 4b). Remarkably, the temperature evolution of the AHC matches very well with the evolution of the FM ordered volume fraction (Fig. 4b, c). The magnitude of the AHC is robust against the increase in temperature up until $T \sim 90$ K, after which it gradually decreases, corresponding to the temperature at which the in-plane AFM volume fraction begins to increase. This is, to our knowledge, the first example of such an excellent correlation between the magnetic volume fraction and the Berry curvature induced AHC. Thus, the direct quantitative link between the magnetic and topological electronic properties is demonstrated. From symmetry arguments, we can extract important clues of which components of the conductivity tensor $\sigma$ are forbidden and which are allowed to be non-zero. Due to the 3-fold rotational symmetry the $\sigma_{xz}$ and $\sigma_{yz}$ components are zero in all considered magnetic structures. The in-plane AFM R-3m structure, shown in Fig. 3e, has a 2-fold rotational symmetry with an axis in the $ab$ plane. Such a rotation transforms $\sigma_{xy}$ to $-\sigma_{xy}$, and hence forces it to be zero (Fig. 4d). On the other hand, for the $c$-axis R-3m′ FM ordering, shown in Fig. 1a, we obtain an extremely high $\sigma_{xy} = 10^3\,\Omega^{-1}\,cm^{-1}$ (Fig. 4d). We can thus conclude from first-principles calculations that the AHC is dominated by the FM $c$-axis component, providing an explanation for the reduction of the AHC when the ordered volume fraction of the out-of-plane FM state decreases. These observations demonstrate for the first time volume-wise magnetic competition

in $Co_3Sn_2S_2$ as well as a remarkable correlation between the magnetically ordered fraction and the AHC. We would like to stress that without appearance of the AFM state, the anomalous Hall conductivity would not depend on temperature within the ordered state. As we have shown here, the anomalous Hall conductivity scales with the FM volume fraction and not with the ordered moment size (see the Supplementary Note 9). At the same time we have shown that the total magnetic fraction (FM + AFM) is constant all the way up to $T_{C2} \simeq 172$ K above which it sharply goes to zero. If the AFM state would not appear above $T_C^* \sim 90$ K, then the FM state would acquire 100% fraction up to 172 K. In this case anomalous Hall conductivity would also exhibit a constant and high value up to 172 K, above which it would sharply go to zero. The reason that anomalous Hall conductivity acquires the temperature dependence above 90 K is strictly due to the appearance of the AFM state. While we did not compute it explicitly, there is no reason to assume that the AFM phase does not support Weyl points. However, the Weyl points would need to be distributed such that the anomalous Hall conductivity vanishes.

**Temperature–pressure and temperature–field-phase diagrams.** For further insight into the magnetic order and the magnetic phase transition in $Co_3Sn_2S_2$, ZF $\mu$SR experiments were carried out as a function of hydrostatic pressure and magnetic field. From the pressure dependent data (see the Supplementary Note 4), we can construct a temperature–pressure phase diagram for $Co_3Sn_2S_2$. The pressure dependences of the transition temperatures $T_{C1}$, $T_{C2}$ and $T_C^*$ are shown in Fig. 5a. We find that hydrostatic pressure has a significant effect on the magnetic properties of this material. Namely, increased pressure results in a substantial reduction of $T_{C1}$, $T_{C2}$ and $T_C^*$ while the fully magnetically ordered volume fraction remains intact. We also find that

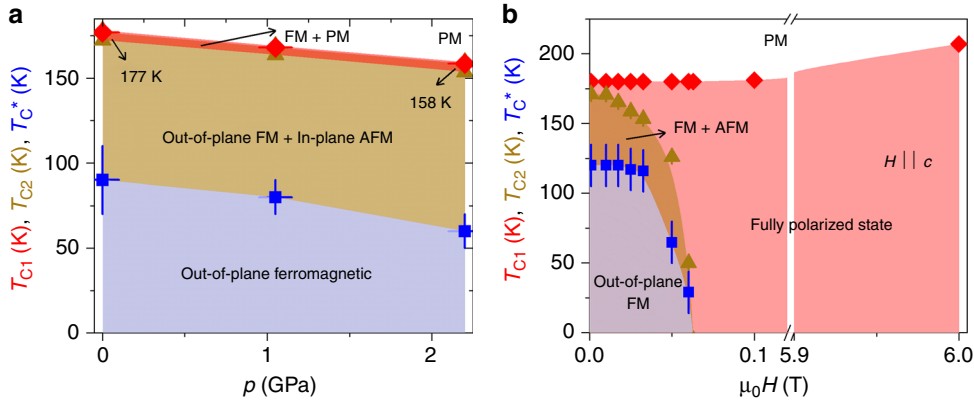

**Fig. 5 Magnetic field and hydrostatic pressure tuning of the magnetic competition in Co₃Sn₂S₂. a** Temperature–pressure phase diagram for $Co_3Sn_2S_2$ obtained by microscopic μSR technique. (**b**) Temperature–field phase diagram for $Co_3Sn_2S_2$, obtained from the macroscopic magnetisation measurements. The error bars represent the s.d. of the fit parameters.

under pressure, the low-frequency component arising from the AFM phase appears at a lower temperature in comparison to ambient pressure. This implies that increased pressure tends to stabilise the high temperature in-plane AFM structure. These findings show that one can physically tune the magnetism in these materials with pressure. We also constructed a temperature–field phase diagram for $Co_3Sn_2S_2$ based on the bulk magnetization data, where the anomalies related to the ferromagnetic and the magnetic transitions are clearly observed (see Supplementary Notes 2 and 3). Figure 5b shows the field dependences of the critical temperatures $T_{C1}$, $T_{C2}$ and $T_C^*$. The phase diagram implies that relatively low fields are enough to tune the magnetic state in $Co_3Sn_2S_2$. The above results show that one can tune the competition between FM and AFM states and have control of the AHC by varying the temperature or varying a non-thermal parameter such as pressure and field.

## Discussion

The exploration of topological electronic phases that result from strong electronic correlations is a frontier in condensed matter physics. Kagome lattice systems are an ideal setting in which strongly correlated topological electronic states may emerge. The simplest Bloch band structure on the kagome lattice naturally includes a flat band[1], in which the effects of electronic correlations are enhanced. Our key finding is to establish $Co_3Sn_2S_2$ as a material that hosts topological electronic states and frustrated magnetism. Our experiments suggest that the Co spins have both ferromagnetic interactions along c-axis and antiferromagnetic interactions within the kagome plane, and there is a temperature-dependent competition between these two ordering tendencies. The interplay between this intricate magnetism and the spin-orbit coupled band structure further induces non-trivial variations of its topological properties, which is characterized by a striking correlation between the anomalous Hall conductivity and the ferromagnetic volume fraction. Our results demonstrate thermal and quantum tuning of anomalous hall conductivity mediated by changes in the frustrated magnetic structure.

## Methods

**General remarks.** We concentrate on the high resolution, high field and high pressure[19–22] muon-spin relaxation/rotation (μSR) measurements of the temperature, pressure and field dependence of the magnetic moment as well as the magnetically ordered volume fraction in $Co_3Sn_2S_2$ and the high-resolution neutron powder diffraction measurements of the tentative magnetic structure at the base temperature. Muon stopping site and local field calculations are carried out to gain insights into the μSR results. In a μSR experiment, positive muons implanted into a sample serve as an extremely sensitive local probe to detect small internal magnetic fields and ordered magnetic volume fractions in the bulk of magnetic materials.

Angle-resolved photoemission spectroscopy (ARPES) measurements and density functional theory calculations were used to explore the electronic band structure. Neutron diffraction has the ability to directly measure the magnetic structure. The techniques of μSR, ARPES and DFT complement each other ideally as we are able to study the detailed temperature dependence of the magnetic order parameter and ordered volume fractions with μSR experiments, and correlate them with the electronic structure measured and calculated by ARPES and DFT, respectively.

**Sample preparation.** Stoichiometric single crystals have been grown by a modified vertical Bridgman technique as described elsewhere[27]. The corresponding polycrystalline $Co_3Sn_2S_2$ were prepared by the solid state reaction method. A stoichiometric ratio of Co (2N+), Sn (2N8) and S (5N) was weighted carefully mixed, and pressed into a pellet. The sample was then evacuated and sealed in double wall quartz ampoule. The sample was slowly heated, with a rate 15 °C/h, up to 450 °C and pre-annealed for 24 h. Subsequently, the sample was heated up to 700 °C, kept at this temperature for 72 h and cooled down to room temperature with a rate 400 °C/h. The whole procedure, with exception of pre-heating, was repeated three times with intermediate re-grounding and pelletizing in a glovebox. To avoid any contamination from the environment the synthesized materials were handled in the helium filled glovebox.

**Pressure cell.** Pressures up to 2.2 GPa were generated in a double wall piston-cylinder type of cell made of CuBe/MP35N material, especially designed to perform μSR experiments under pressure[19–22]. A pressure transmitting medium Daphne oil was used. The pressure was measured by tracking the SC transition of a very small indium plate by AC susceptibility. The filling factor of the pressure cell was maximized. The fraction of the muons stopping in the sample was ~40%.

**Neutron powder diffraction experiments.** The magnetic and crystal structures in $Co_3Sn_2S_2$ have been studied by neutron powder diffraction (NPD) in the temperature range 2–250 K. The NPD experiments were carried out at the Swiss Spallation Neutron Source (SINQ), Paul Scherrer Institute, Villigen, Switzerland. Approximately 2.5 g of powder sample was loaded into a 6-mm diameter vanadium can. The diffraction patterns were recorded between 1 and 250 K at the cold neutron powder diffractometer (DMC)[28] using $\lambda = 2.4576$ Å (pyrolytic graphite (002), $2\theta_{max} = 92.7°$, $2\theta_{step} = 0.1°$) and at the High-Resolution Powder diffractometer for Thermal neutrons (HRPT)[29] using $\lambda = 1.154$, 1.8857 and 2.449 Å [Ge (822), $2\theta_{max} = 160°$, $2\theta_{step} = 0.05°$]. Large statistics acquisitions for magnetic structure refinements were made at both 2 and 170 K. The refinements of the crystal structure parameters were done using FullProf suite[30], with the use of its internal tables for neutron scattering lengths. The symmetry analysis was performed using ISODISTORT tool based on ISOTROPY software[31,32], BasiRep program[30] and software tools of the Bilbao crystallographic server[33].

**μSR experiment.** In a μSR experiment nearly 100% spin-polarized muons ($\mu^+$) are implanted into the sample one at a time. The positively charged $\mu^+$ thermalize at interstitial lattice sites, where they act as magnetic microprobes. In a magnetic material the muon-spin precesses in the local field $B_\mu$ at the muon site with the Larmor frequency $\nu_\mu = \gamma_\mu/(2\pi)B_\mu$ [muon gyromagnetic ratio $\gamma_\mu/(2\pi) = 135.5$ MHz T$^{-1}$].

The low background GPS ($\pi$M3 beamline) instrument was used to study the single crystalline as well as the polycrystalline samples of $Co_3Sn_2S_2$ at ambient pressure. μSR experiments under pressure were performed at the μE1 beamline of the Paul Scherrer Institute (Villigen, Switzerland), where an intense high-energy ($p_\mu = 100$ MeV/c) beam of muons is implanted in the sample through the pressure

cell. Transverse-field (TF) μSR experiments were performed at the πE3 beamline of the Paul Scherrer Institute (Villigen, Switzerland), using the HAL-9500 μSR spectrometer. The specimen was mounted in a He gas-flow cryostat with the c-axis parallel to the muon beam direction, along which the external field was applied. Magnetic fields between 10 mT and 8 T were applied, and the temperatures were varied between 3 and 300 K.

**Analysis of weak TF-μSR data.** The wTF asymmetry spectra were analysed by the function[34]:

$$A_S(t) = A_p \exp(-\lambda t) \cos(\omega t + \phi),\qquad(1)$$

where t is time after muon implantation, A(t) is the time-dependent asymmetry, $A_p$ is the amplitude of the oscillating component (related to the paramagnetic volume fraction), λ is an exponential damping rate due to paramagnetic spin fluctuations and/or nuclear dipolar moments, ω is the Larmor precession frequency set, which is proportional to the strength of the transverse magnetic field, and ϕ is a phase offset. As it is standard for the analysis of wTF data from magnetic samples the zero for A(t) was allowed to vary for each temperature to deal with the asymmetry baseline shift known to occur for magnetically ordered samples. From these refinements, the magnetically ordered volume fraction at each temperature T was determined by $1 - A_p(T)/A_p(T_{max})$, where $A_p(T_{max})$ is the amplitude in the paramagnetic phase at high temperature.

**Analysis of ZF-μSR data.** All of the ZF-μSR spectra were fitted using the following model[34]:

$$A_{ZF}(t) = F\left(\sum_{j=1}^{2}\left(f_j \cos(2\pi\nu_j t + \phi)e^{-\lambda_j t}\right) + f_L e^{-\lambda_L t}\right) + (1-F)\left(\frac{1}{3} + \frac{2}{3}\left(1 - (\sigma t)^2\right)e^{-\frac{1}{2}(\sigma t)^2}\right).\qquad(2)$$

The model (2) consists of an anisotropic magnetic contribution characterized by an oscillating transverse component and a slowly relaxing longitudinal component. The longitudinal component arises due to the parallel orientation of the muon-spin polarization and the local magnetic field. In polycrystalline samples with randomly oriented fields, this results in a so-called one-third tail with $f_L = \frac{1}{3}$. For single crystals, $f_L$ varies between zero and unity as the orientation between field and polarization changes from being parallel to perpendicular. Note that the whole volume of the sample is magnetically ordered nearly up to $T_C$. Only very near the transition, there is a PM signal component, in addition to the magnetically ordered contribution, that is characterized by the densely distributed network of nuclear dipolar moments with a corresponding relaxation rate σ. The temperature-dependent fraction $0 \le F \le 1$ governs the trade-off between magnetically ordered and PM behaviours.

**Analysis of ZF-μSR data under pressure.** In the pressure dependent experiments, a substantial fraction of the μSR asymmetry originates from muons stopping in the MP35N pressure cell surrounding the sample. Therefore, the μSR data in the entire temperature range were analysed by decomposing the signal into a contribution of the sample and a contribution of the pressure cell:

$$A(t) = A_S(0)P_S(t) + A_{PC}(0)P_{PC}(t),\qquad(3)$$

where $A_S(0)$ and $A_{PC}(0)$ are the initial asymmetries and $P_S(t)$ and $P_{PC}(t)$ are the normalized muon-spin polarizations belonging to the sample and the pressure cell, respectively. The pressure cell signal was analysed by a damped Kubo-Toyabe function[21].

**DFT calculations.** Fully-relativistic band structure calculations were performed within the VASP[35,36] code employing the projector-augmented wave method (PAW)[37,38] and the Perdew, Burke, and Ernzerhof generalized-gradient (GGA-PBE) exchange-corellation functional[39]. The AHC was calculated by means of the Wannier interpolation scheme[40], as implemented in the Wannier90 package[41,42]. The starting projections for constructing maximally-localized Wannier functions[43] were chosen as sp3 orbitals of Sn and S and the d-orbitals of Co. The upper bound for the inner and outer window for the band disentanglement procedure[44] were chosen at +2 and +5 eV relative to the Fermi level. The surface density of states was calculated for the Sn-terminated surface on the basis of the tight binding model derived from Wannier functions, by means of the WannierTools package[45].

## Data availability

All relevant data are available from the authors. The data can also be found at the following link http://musruser.psi.ch/cgi-bin/SearchDB.cgi.

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

## Acknowledgements

The μSR experiments were carried out at the Swiss Muon Source (SμS) Paul Scherrer Insitute, Villigen, Switzerland, using the high field HAL-9500 μSR spectrometer (πE3 beamline), GPS instrument (πM3 beamline) and high pressure GPD instrument (μE1 beamline). The neutron diffraction experiments were performed at the Swiss spallation neutron source SINQ (HRPT and DMC diffractometers), Paul Scherrer Institute, Villigen, Switzerland. Z.G. thanks Vladimir Pomjakushin for invaluable support with neutron diffraction experiments/analysis and for his useful discussions. H.C.L. thanks the support by the National Key RandD Program of China (Grants No. 2016YFA0300504), the National Natural Science Foundation of China (No. 11574394, 11774423, 11822412). This project has received funding from the European Research Council (ERC) under the European Union's Horizon 2020 research and innovation programm (ERC-StG-Neupert-757867-PARATOP). The magnetization measurements were carried out on the PPMS/MPMS devices of the Laboratory for Multiscale Materials Experiments, Paul Scherrer Institute, Villigen, Switzerland. This work was supported the Swiss National Science Foundation (grant no. 206021_139082). Work at Princeton University is supported by the Gordon and Betty Moore Foundation (GBMF4547/M.Z.H.)and the U.S. Department of Energy (DOE) under Basic Energy Sciences (DOE/BES DE-FG-02-05ER46200). M.Z.H. acknowledges support from the Miller Institute of Basic Research in Science at the University of California at Berkeley and Lawrence Berkeley National Laboratory in the form of a Visiting Miller Professorship during the early stages of this work. M.Z.H. also acknowledges visiting scientist support from IQIM at the California Institute of Technology.

## Author contributions

M.Z.H. and Z.G. conceived the study. M.Z.H. supervised the project. μSR experiments and corresponding data analysis: Z.G., J.V., G.S., H.L., R.K. and A.A. Neutron diffraction experiments: Z.G., D.G. and L.K. Magnetization measurements: Z.G. and D.G. STM experiments: J.-X.Y. and S.-S.Z. in consultation with M.Z.H. ARPES experiments: I.B., Z.G. and T.A.C. in consultation with M.Z.H. Growth of single crystals: H.Z., Q.W., H.C.L. and S.J. Conductivity measurements: H.Z. and S.J. Growth of polycrystalline samples: D.G., Z.S. and E.P. Band structure and AHC calculations: S.S.T., G.C, and T.N. Writing the paper: Z.G., J.-X.Y. and M.Z.H. with contributions from all authors. All authors discussed the results, interpretation and conclusion.

## Competing interests

The authors declare no competing interests.
