## [Peer Review File · Nature Communications]

Reviewers' comments:

Reviewer #2 (Remarks to the Author):

In the previous review for Nature Materials, I argued that the Berry curvature can be varied at elevated temperatures even without the development of antiferromagnetism, while it may accelerate the variation. The authors disagreed with this remark in the reply and insisted that volume-wise antiferromagnetism is required to tune the Berry curvature of the band structure. The related statements are found in the main texts as followings:

"(main text) We note that the change in the AHC represents a change in the Berry curvature of the occupied states. The change comes from the modifications of the band dispersion or of the Berry curvature of the occupied states themselves. Thus, these findings provide us with the unique possibility of a thermal or quantum tuning of Berry curvature in Co₃Sn₂S₂ through tuning the magnetic competition."

I wonder if these statements are proper. In the review comments, I raised my concerns as follows:

"(my comments) I am also wondering if Berry curvature is really tuned by the magnetic competition when these two states are spatially distributed, and each magnetic domain forms a band structure: the ferromagnetic domains have Weyl points, whereas the antiferromagnetic domains don't. In that case, one cannot say that the Berry curvature is tunable since the signature of anomalous Hall conductivity should be just the summation coming from spatially distributed states, not due to the Berry curvature varied in one band. The author should discuss this issue."

Replying to these, the authors changed the title with a more moderate expression:

"Tunable Berry Curvature Effects Through Volume-wise Magnetic Competition in a Topological Kagome Magnet Co₃Sn₂S₂"

I agree that this title is more relevant to the findings by the authors. In the reply letters, the authors also seem to carefully use phrases as in the following statements: "the thermal evolution of Berry curvature induced anomalous Hall effect" without more strongly expressing like "the change in the AHC represents a change in the Berry curvature of the occupied states."

The careful stance of the authors is actually seen in the following reply by the authors:

"(reply by the authors) While we did not compute it explicitly, there is no reason to assume that the AFM phase does not support Weyl points. However, the Weyl points would need to be distributed such that the anomalous Hall conductivity vanishes."

I suggest that the authors add these statements in the main text, and avoid mentioning "the change in the AHC represents a change in the Berry curvature of the occupied states" since there is no such evidence in the data presented in this manuscript. The data are shown in this manuscript mostly indicate that the Hall voltage observed is reduced with temperature simply because the volume of AF domain, which generates Hall voltage, is reduced because of the development of AFM domains, which do not generate Hall voltage. The total Hall voltage along the sample should be the spatial summation of Hall voltage for the two domains. The Berry curvature for AF domain (AFM domain) should be the same no matter how the AFM domain (AF domain) is developed. In addition, I think that one cannot define a band structure for the summation of two domains since the AF domain and the AFM domain has each different band structure. Therefore, I don't think the following (or similar) statement in the main text is proper: "AHC represents a change in the Berry curvature of the occupied states". From the data, it is not clear whether the

Berry curvature for the AF domain can be changed by the development of AFM domain, or it is changed simply with temperature, or it is changed neither by the development of AFM domain nor by temperature.

In conclusion, I cannot accept the manuscript unless the above argument is properly addressed.

Reviewer #3 (Remarks to the Author):

The authors have addressed most of my comments in the revised version of their manuscript. I have to admit this work does not clarify the physical mechanism underlying the competition between the two magnetically phases. Nevertheless, the experimental data seem to provide a compelling evidence such a competition is indeed realised in this interesting materials. At this point, I would not object against publishing this work in Nature Communications.

Manuscript: Tunable Anomalous Hall Conductivity Through Volume-wise Magnetic Competition in a Topological Kagome Magnet $\text{Co}_3\text{Sn}_2\text{S}_2$

Reply to the Referee 2

In the previous review for Nature Materials, I argued that the Berry curvature can be varied at elevated temperatures even without the development of antiferromagnetism, while it may accelerate the variation. The authors disagreed with this remark in the reply and insisted that volume-wise antiferromagnetism is required to tune the Berry curvature of the band structure. The related statements are found in the main texts as followings.

”(main text) We note that the change in the AHC represents a change in the Berry curvature of the occupied states. The change comes from the modifications of the band dispersion or of the Berry curvature of the occupied states themselves. Thus, these findings provide us with the unique possibility of a thermal or quantum tuning of Berry curvature in $\text{Co}_3\text{Sn}_2\text{S}_2$ through tuning the magnetic competition.”

I wonder if these statements are proper. In the review comments, I raised my concerns as follows:

”(my comments) I am also wondering if Berry curvature is really tuned by the magnetic competition when these two states are spatially distributed, and each magnetic domain forms a band structure: the ferromagnetic domains have Weyl points, whereas the antiferromagnetic domains don't. In that case, one cannot say that the Berry curvature is tunable since the signature of anomalous Hall conductivity should be just the summation coming from spatially distributed states, not due to the Berry curvature varied in one band. The author should discuss this issue.”

Replying to these, the authors changed the title with a more moderate expression: ”Tunable Berry Curvature Effects Through Volume-wise Magnetic Competition in a Topological Kagome Magnet $\text{Co}_3\text{Sn}_2\text{S}_2$ ”.

I agree that this title is more relevant to the findings by the authors. In the reply letters, the authors also seem to carefully use phrases as in the following

statements: "the thermal evolution of Berry curvature induced anomalous Hall effect" without more strongly expressing like "the change in the AHC represents a change in the Berry curvature of the occupied states."

The careful stance of the authors is actually seen in the following reply by the authors: "(reply by the authors) While we did not compute it explicitly, there is no reason to assume that the AFM phase does not support Weyl points. However, the Weyl points would need to be distributed such that the anomalous Hall conductivity vanishes."

I suggest that the authors add these statements in the main text, and avoid mentioning "the change in the AHC represents a change in the Berry curvature of the occupied states" since there is no such evidence in the data presented in this manuscript. The data are shown in this manuscript mostly indicate that the Hall voltage observed is reduced with temperature simply because the volume of AF domain, which generates Hall voltage, is reduced because of the development of AFM domains, which do not generate Hall voltage. The total Hall voltage along the sample should be the spatial summation of Hall voltage for the two domains. The Berry curvature for AF domain (AFM domain) should be the same no matter how the AFM domain (AF domain) is developed. In addition, I think that one cannot define a band structure for the summation of two domains since the AF domain and the AFM domain has each different band structure. Therefore, I don't think the following (or similar) statement in the main text is proper: "AHC represents a change in the Berry curvature of the occupied states". From the data, it is not clear whether the Berry curvature for the AF domain can be changed by the development of AFM domain, or it is changed simply with temperature, or it is changed neither by the development of AFM domain nor by temperature.

In conclusion, I cannot accept the manuscript unless the above argument is properly addressed.

We are very grateful to the Referee 2 for carefully studying the manuscript and providing us with the useful comments. We fully agree with the Referee 2 that it is too speculative to say that the Berry curvature is tuned by the magnetic competition. What we measure experimentally is the anomalous hall effect and not Berry curvature. Thus, according to his/her suggestion, we fully avoided to claim that Berry curvature is tuned by temperature and

removed the paragraph, stating the following “the change in the AHC represents a change in the Berry curvature of the occupied states“. Instead, we make statements, which accurately reflect our experimental findings: the appearance of the AFM phase at high temperatures as well as the volume wise magnetic competition between FM and AFM phases and the striking correlation between the FM volume fraction and the anomalous hall conductivity. This implies that the magnetic competition drives the thermal evolution of the anomalous hall conductivity. Within the AFM phase, the Weyl points would need to be distributed such that the anomalous Hall conductivity vanishes. Led by the Referee’s comments, we also changed the title to the following: Tunable Anomalous Hall Conductivity Through Volume-wise Magnetic Competition in a Topological Kagome Magnet $\text{Co}_3\text{Sn}_2\text{S}_2$.

Reply to the Referee 3

The authors have addressed most of my comments in the revised version of their manuscript. I have to admit this work does not clarify the physical mechanism underlying the competition between the two magnetically phases. Nevertheless, the experimental data seem to provide a compelling evidence such a competition is indeed realised in this interesting materials. At this point, I would not object against publishing this work in Nature Communications.

We would like to thank Referee 3 for very positive report and supporting the acceptance of the paper in Nature Communications.

REVIEWERS' COMMENTS:

Reviewer #2 (Remarks to the Author):

In the previous review, I did not agree with the authors' arguments that the Berry curvature is tuned by the magnetic competition and pointed out that the anomalous Hall conductivity decreases because the volume fraction of FM domains generating it is reduced by the development of the competing AF domains.

In the revised manuscript, I could find that the authors carefully corrected confusing statements, and now it is clear that the main result of this paper is "Tunable Anomalous Hall Conductivity Through Volume-wise Magnetic Competition", which is the title of the revised manuscript and is clearly demonstrated by their data. This result is still intriguing and can expect a high impact on the community, thus I can now agree that the current version of the manuscript is suitable as a paper of Nature communications.

Manuscript: Tunable Anomalous Hall Conductivity Through Volume-wise Magnetic Competition in a Topological Kagome Magnet $\text{Co}_3\text{Sn}_2\text{S}_2$

Reply to the Reviewer 2

In the previous review, I did not agree with the authors' arguments that the Berry curvature is tuned by the magnetic competition and pointed out that the anomalous Hall conductivity decreases because the volume fraction of FM domains generating it is reduced by the development of the competing AF domains.

In the revised manuscript, I could find that the authors carefully corrected confusing statements, and now it is clear that the main result of this paper is "Tunable Anomalous Hall Conductivity Through Volume-wise Magnetic Competition", which is the title of the revised manuscript and is clearly demonstrated by their data. This result is still intriguing and can expect a high impact on the community, thus I can now agree that the current version of the manuscript is suitable as a paper of Nature communications.

We would like to thank Reviewer 2 for very positive report and supporting the acceptance of the paper in Nature Communications.